# Effects of Sweet and Forge Sorghum Silages Compared to Maize Silage without Additional Grain Supplement on Lactation Performance and Digestibility of Lactating Dairy Cows

**DOI:** 10.3390/ani14111702

**Published:** 2024-06-05

**Authors:** Sujiang Zhang, Jiao Wang, Shunping Lu, Abdul Shakoor Chaudhry, Divine Tarla, Hassan Khanaki, Imtiaz Hussain Raja, Anshan Shan

**Affiliations:** 1Key Laboratory of Tarim Animal Husbandry Science and Technology, College of Animal Science and Technology, Tarim University, Alar 843300, China; 18119223476@163.com; 2Key Laboratory of Livestock and Grass Resources Utilization around Tarim, Ministry of Agriculture and Rural Areas (Co-Construction by Ministries and Provinces), Tarim University, Alar 843300, China; 3College of Life Sciences and Technology, Tarim University, Alar 843300, China; wang20220910@126.com; 4School of Natural and Environmental Sciences, Newcastle University, Newcastle upon Tyne NE1 7RU, UK; abdul.chaudhry@newcastle.ac.uk; 5School of Agriculture, Food and Ecosystem Sciences, Faculty of Science, The University of Melbourne, Dookie College, VIC 3647, Australia; d.ngwatarla@unimelb.edu.au (D.T.); h.khanaki@unimelb.edu.au (H.K.); 6Department of Animal Nutrition, Faculty of Animal Production and Technology, Cholistan University of Veterinary and Animal Sciences, Bahawalpur 63100, Pakistan; imtiazhussain@cuvas.edu.pk; 7Institute of Animal Nutrition, Northeast Agricultural University, Harbin 150030, China; asshan@neau.edu.cn

**Keywords:** Holstein cows, maize silage, milk, nitrogen use efficiency, high sugar sorghum

## Abstract

**Simple Summary:**

Maize silage is one of the most commonly used forages on many dairy farms. However, the absence of tiller and regeneration characteristics, alongside its high-water needs, limit its planting potential in arid areas. Conversely, the cultivation of sweet sorghum for silage in arid regions worldwide has been constantly increasing due to its excellent regrowth, tiller, biomass yield, water-soluble carbohydrates, and resistance to drought. Our previous in vitro studies have shown that sweet sorghum can replace maize in arid areas to produce high-quality silage feed. This study assumes that the sugar in sweet sorghum silage can provide the required proportion of starch content. Therefore, this study evaluated the effects of high-sugar sorghum silage, forage sorghum silage, and maize silage on the lactation performance and digestibility of dairy cows. It was observed that feeding sweet sorghum silage without additional grain supplementation was substantially similar to feeding maize silage. No differences in milk yield or nutrient digestibility in cows were noted for these tested forages. This indicates that sweet sorghum silage could be an acceptable feedstuff to support milk production in dairy cattle, especially in water-limited regions worldwide.

**Abstract:**

This study investigated the effects of replacing maize silage (MZS) with high-sugar sorghum silage (HSS) or forage sorghum silage (FSS) without additional grain supplement in the diets of dairy cows on nutrient digestibility, milk composition, nitrogen (N) use, and rumen fermentation. Twenty-four Chinese Holstein cows (545 ± 42.8 kg; 21.41 ± 0.62 kg milk yield; 150 ± 5.6 days in milk) were randomly assigned to three dietary treatments *(n* = 8 cows/treatment). The cows were fed ad libitum total mixed rations containing (dry matter basis) either 40% MZS (MZS-based diet), 40% HSS (HSS-based diet), or 40% FSS (FSS-based diet). The study lasted for 42 days, with 14 days devoted to adaptation, 21 days to daily feed intake and milk production, and 7 days to the sampling of feed, refusals, feces, urine, and rumen fluid. Milk production was measured twice daily, and digestibility was estimated using the method of acid-insoluble ash. The data were analyzed using a one-way ANOVA in SPSS 22.0 according to a completely randomized design. Dietary treatments were used as fixed effects and cows as random effects. The results indicate that MZS and HSS had greater crude protein but less neutral detergent fiber (NDF), acid detergent fiber (ADF), acid detergent lignin (ADL), and a lower pH than FSS (*p* ≤ 0.04). High starch contents in MZS and water-soluble carbohydrate (WSC) contents in HSS were observed (*p* < 0.01). While the highest starch intake was observed for the MZS-based diet, the highest WSC intake was noted for the HSS-based diet, and the highest NDF, ADF, ADL intake was observed for the FSS-based diet (*p* ≤ 0.05). The diets, including MZS and HSS, had greater digestibility than that of FSS (*p* ≤ 0.03). Feeding MZS- and HSS-based diets increased the yield, fat, and protein content of the milk, as well as feed conversion efficiency (*p* ≤ 0.03). However, feeding the MZS- and HSS-based diets decreased the contents of milk urea N, urinary urea N, and urinary N excretion more than the FSS-based diet (*p* ≤ 0.05). The N use efficiency tended to increase relative to diets containing MZS and HSS compared with FSS (*p* = 0.06 and *p* = 0.09). Ruminal ammonia-N and pH were lower, but total volatile fatty acids, acetate, and propionate were higher in cows fed the HSS- and MZS-based diets compared to those fed the FSS-based diet (*p* ≤ 0.03). It appears as though replacing MZS with HSS in the diet of cows without additional grain supplements has no negative influence on feed intake, milk yield, N utilization, or ruminal fermentation.

## 1. Introduction

Due to more undeveloped land, less environmental pollution, easier epidemic prevention, and lower construction costs of livestock farms, more cattle farms are built in arid and semiarid areas around the world. Although the dry environment is relatively good for cattle breeding, it is detrimental for forage growth [1]. The capacity of providing natural roughage from arid and semiarid lands has declined remarkably, due to reclaiming wasteland, water shortage, and severe soil salinity [2]. Recently, the shortage of roughage has become the main factor limiting the increasing scale of raising cattle in arid and semiarid regions.

In arid and semiarid regions, maize (*Zea mays* L.) is a principal forage and is commonly cultivated for silage (MZS). However, its yield and quality can be negatively affected by groundwater shortages, limited precipitation, and severe salinity in these regions [3]. Additionally, maize requires more fertilizer compared to sweet sorghum [4,5], a fact which can increase production costs. Therefore, exploring alternative water-efficient plants such as sweet sorghum (*Sorghum bicolor* L. Moench), which, with a root depth of up to 270 cm, can fully utilize 0–91 cm of water [5], to replace maize for forage production is needed in arid and semiarid areas [6].

Sweet sorghum has been increasingly used as a substitute for maize to produce silage in ruminant feeding [3,7] owing to its excellent biological characteristics, such as high photosynthetic efficiency and biomass yield. Indeed, sweet sorghum, under severe drought stress, can produce 27% more aboveground dry matter (DM) than maize. Schittenhelm and Schroetter [8] reported greater content of water-soluble carbohydrates (WSC, sweet sorghum 140–484 g/kg vs. maize 122 g/kg [6]), good regrowth (sweet sorghum can be harvested twice vs. maize which can only be harvested once per year), strong drought resistance [9,10,11], and low fertilizer requirements (sweet sorghum required just 36% of the nitrogen (N) fertilizer needed for maize [5]). Sweet sorghum can be classified as either a high-sugar sweet sorghum or forage sorghum, depending on its sugar content. Its silage is mostly for feeding ruminants. However, the effects of sweet sorghum silage on milk yield and composition, reported in different studies, are inconsistent. Some studies have indicated that the replacement of MZS with sorghum silage in dairy cow diets reduced neither the actual or fat-corrected milk (FCM) yield nor milk components [12,13,14], while other studies have reported contradictory results [15,16,17]. In these studies, the substitution of MZS with either forage sorghum (hybrid Hannibal) or high-sugar sweet sorghum (BJ0603) silages was accompanied by supplementing grains in sorghum-silage-based diets to reduce the fiber content and compensate for the lower starch concentration compared with the MZS-based diets [14,18]. In production, breeders also suspect that the effect of raising cows with sweet sorghum silage is not as good as that of raising cows with maize silage. However, our previous study showed that high-sugar sorghum could be an alternative to maize for making good-quality silage in arid regions, based on the results from chemical analyses and an in vitro fermentation study [6]. These in vitro results needed to be further confirmed by conducting an animal experiment.

It is hypothesized that the sugar in sweet sorghum silages can provide the required proportion of starch, which can maintain the milk production performance of dairy cows and improve the efficiency of N utilization. Thus, the present research explores the effects of high-sugar sorghum silage (HSS), forage sorghum silage (FSS), and MZS on milk yield, nutrient digestibility, and N metabolism in Chinese Holstein cows.

## 2. Materials and Methods

The procedures used in the study were conducted in accordance with the Standards for the Care and Use of Animals for Research in China (GB 14925-2001) [19]. The experimental protocol (No. 2018023) used in this study was approved by the Animal Research Ethics Committee of Tarim University, and the study was performed on a dairy farm (Jiangnan Husbandry Co., Ltd., Tumushuke) in the Xinjiang province of China.

### 2.1. Crops Cultivation and Silage Making

The high-sugar sorghum, forage sorghum, and maize crops were planted on adjacent fields at Jiangnan Husbandry Farm, located in Tumushuke (longitude 79°06′ E, latitude 39°86′ N), Xinjiang Province, China. This is a temperate and extremely arid desert region with mean annual precipitations of 38.3 mm and average temperatures of 11.6 °C. The average temperature in May was 22.5 °C, without precipitation during crop cultivation. The cultivars Rio (22.1% brix) as high-sugar sorghum, X096 (9.4% brix) as forage sorghum, and BY02 as maize were used in this study. Basal fertilizers of 35 kg N/ha as urea, 70 kg P_2_O_5_/ha as diammonium phosphate, and 40 kg K_2_O/ha as potassium sulfate were applied just before sowing. A planter (2BMQYF-2; Shandong Denong Agricultural Machinery Manufacturing Co., Ltd., Dezhou, China) was used for sowing at a rate of 15 kg/ha for both high-sugar sorghum and forage sorghum and 40 kg/ha for maize. For weed control, 2.5 L of atrazine/ha was sprayed before tillage and a week after germination. The crops were watered six times during the growth period through drip irrigation. All cultural practices applied were the same on all plots. About five months post-sowing, all the plants were harvested whole using a self-propelled forage chopper (4YZ-4B, Xinxiang Kaihang Machinery Manufacturing Co., Ltd., Xinxiang, China) at the early dough stage with about 70% water content. They were chopped at a theoretical chop length of 2.5 cm and respectively ensiled in separate underground silos (each cuboid silo was 15 m long, 4 m wide, and 1.5 m high) to produce HSS, FSS, and MZS without adding inoculants. The materials were compacted by driving tractors over them. The silos were kept sealed until the beginning of the feeding study.

About 15 g of fresh sample from each experimental silage was mixed with 135 mL of distilled water, and the mixture was left for 1 h at 25 °C with occasional stirring, followed by filtration through two layers of cheesecloth. The filtrate was then tested for pH using a digital pH meter (PHS-2F, Oustor Industrial Co., Ltd., Shanghai, China). About 500 g of fresh silage samples was taken from different points of individual silages. The silage samples were oven-dried at 65 °C to a constant weight and ground to pass a 1 mm sieve for chemical analysis.

### 2.2. Experimental Diets and Animals

Three experimental diets (Table 1) containing MZS, HSS, or FSS were formulated as total mixed rations using a mixing wagon (Henan Haomu Machinery Co., Ltd., Zhengzhou, China). The chemical compositions of experimental diets are the measured value, while NFC (non-fiber carbohydrates) and NE_L_ (net energy for lactation) are the calculated values.

Twenty-four healthy multiparous mid-lactation (150 ± 5.6 days in milk) Chinese Holstein cows with similar body weight (545 ± 42.8 kg) and similar milk production (21.41 ± 0.62 kg/day) at the beginning of the study were randomly assigned in equal numbers to the three experimental dietary treatment groups (Table 1), with eight cows in each group according to a completely randomized design. All animals were kept in stalls, which were equipped with feeders and water drinkers. The experimental diets were individually offered ad libitum to the animals twice a day at 08:00 and 19:00, and the offered amounts were adjusted to achieve approximately 10% refusals. Fresh water was freely available to all cows throughout the experiment.

The study lasted for 42 days, with 14 days devoted to the adaptation to feeding and housing routines and 21 days to measuring the daily feed intake and milk production. The amount of diet provided to each experimental cow and corresponding leftovers were recorded daily to estimate voluntary feed intake, which was used for calculating nutrient intake and the efficiency of milk yield for each diet. The last 7 days were used for the collection of samples (i.e., diet, feces, urine, milk, and rumen fluid) and measurement of nutrient digestibility as described below.

### 2.3. Milk Sampling and Analysis

The cows were milked twice a day using a vacuum dairy milking machine (YZ-IIFQ, Zibo Glory Machinery Co., Ltd., Zibo, China). The milk yield for each experimental cow was recorded daily. A yield of 3.5% FCM was computed using the following equation, as described by Holt et al. [21]:3.5% FCM = (0.4324 × kg of milk yield) + (16.216 × kg of milk fat)(1)

About 50 mL of milk samples was collected from each of these cows using milk meters during each milking event over three consecutive days, from day 36 to day 38. Milk samples were preserved with K_2_Cr_2_O_7_ and stored at 4 °C for analysis. True protein, lactose, fat, milk urea N, total solids (TS), and solids-not-fat from milk samples synthesized by each animal were measured using an infrared instrument (UL40BC, Wobeng Instrument Equipment Co. Ltd., Zhengzhou, China) calibrated weekly using raw milk standards. Milk component yields were determined by multiplying milk yield by the concentration of milk composition. Conversion efficiencies of feed-to-milk production were computed as milk yield and 3.5% FCM divided by DM intake.

### 2.4. Nutrient Digestibility and Chemical Analysis

To reduce labor intensity and save time, the total tract apparent digestibility of DM, organic matter (OM), crude protein (CP), neutral detergent fiber (NDF), acid detergent fiber (ADF), acid detergent lignin (ADL), WSC, and starch were estimated using acid-insoluble ash as an internal marker method, as previously described [22]. Briefly, about 200 g of fecal and diet matter was collected, respectively, from day 36 to day 38. Fecal samples were grabbed before feeding, and 10 mL of 10% (vol/vol) sulfuric acid (H_2_SO_4_) was added to the feces to avoid ammonia-N (NH_3_-N) volatilization. The fecal samples collected from each cow were weighed and properly mixed each day, and 1% of each well-mixed feces was picked up as a fecal sample and stored at −20 °C for chemical analysis. The acid-insoluble ash contents of each diet and fecal samples were determined using 4 N HCl, as described by Van Keulen and Young [23]. The equation to estimate the digestibility of each nutrient is as follows [22]:Nutrient digestibility % = (nutrients in diet % × acid-insoluble ash in feces % − acid-insoluble ash in diet% × nutrient in feces %) ÷ (acid-insoluble ash in feces % × nutrient in diet %)(2)

The nutrient levels of DM, CP, NDF, ADF, ADL, WSC, and starch in silages, feeds, and feces were tested. In addition, the ether extract (EE), ash, Ca, and P of feeds, as well as the EE, ash, total phenolic substances, condensed tannins, and pH of silages were also measured. The DM was determined by drying samples at 65 °C for 48 h (AOAC, 934.01), the ash was determined by combustion at 550 °C for 5 h (AOAC, 935.42), and the OM in silages and experimental diets was calculated as the difference between 100 and the ash concentrations of the corresponding samples. The content of EE was determined by a Soxhlet apparatus with petroleum ether as the extraction solvent using the method 989.05 (AOAC, 2016) [24]. The N was analyzed using an automatic Kjeldahl N Determiner (K9840, Hanon Scientific Instruments Co., Ltd., Jinan, China) and CP was determined by multiplying N by 6.25. The NDF contents were analyzed after the addition of heat-stable α-amylase and sodium sulfite, and the ADF and ADL contents were expressed exclusive of ash by a fiber analyzer (F800, Hanon Scientific Instruments Co., Ltd., Jinan, China) using the methods by Van Soest et al. [25]. The WSC contents were measured using a spectrophotometer (V-5600 (PC), Shanghai Metash Instruments Co., Ltd., Shanghai, China) [26]. The starch concentration was determined according to the methods reported by Chaudhry and Khan [27]. For each sample, 2 g was weighed into a funnel with filter paper. Each was washed with 30 mL of ether, 150 mL of alcohol, and 100 mL of distilled water. The washed filter residues were transferred into 250 mL conical flasks and refluxed with 10% HCl. This was then neutralized with 5 mol/L NaOH, diluted with water, and titrated against freshly made Fehling’s solution (69.28 g copper sulphate, 346 g potassium sodium tartrate tetrahydrate, and 120 g sodium hydroxide/L distilled water). Total phenolic substances were analyzed using the Folin–Ciocalteu method [28]. Total condensed tannins were determined using the method reported by Osman [29].

### 2.5. Urine Sampling and Analysis

Spot urine samples were collected using tubes in containers for each cow from day 33 to day 35 before the morning feeding, when cows urinated spontaneously. The urine samples (10 mL) were diluted with 40 mL of 10% (vol/vol) H_2_SO_4_ and frozen at −20 °C until analysis for urea N concentration [21]. The N intake was determined by the concentration of feed N and the amount of feed intake and leftovers. A conversion factor of 6.38 was used to convert milk true protein to milk N [30]. The urinary urea N was measured using ELISA assay kits (Shanghai Jiya Biotechnology Co., Ltd., Shanghai, China) through a spectrophotometer (Thermo Scientific Multiskan SkyHigh, Thermo Fisher Technology Co., Ltd., Shanghai, China). The urinary N excretion was calculated as the N concentration in urine multiplied by the daily urine volume [31]. Total urine was collected daily from cows during the last 3 days of the experiment with the indwelling Folley catheters (24 French, 75 mL balloons) through the urethra, such catheters inserted on day 39 of the experiment. Fecal N excretion was calculated using the equation: N intake − urinary N excretion − milk N. The manure N was computed using the equation of urinary N excretion + fecal N excretion. The N utilization efficiency was expressed by the ratios of milk N to intake N and milk N to manure N.

### 2.6. Rumen Fluid Sampling and Analysis of pH, Ammonia-Nitrogen, and Volatile Fatty Acids

Approximately 200 mL of ruminal fluid was collected daily from each cow at 3 h post-morning feeding from day 35 to day 37 via oral intubation of a stomach tube into the rumen (Kelibo Technology, Wuhan, China). The first 50 mL of ruminal fluid was discarded to prevent saliva contamination. The collected samples were filtered through the double layers of gauze, and the pH was determined using a digital pH meter (PHS-2F, Oustor Industrial Co., Ltd., Shanghai, China). About 0.25 mL of 50% H_2_SO_4_ per 10 mL of ruminal fluid was added to preserve ruminal fluid subsamples for NH_3_-N analysis, and 5 mL formic acid per 5 mL of ruminal fluid was added to preserve subsamples for volatile fatty acids (VFA) analysis. The phenol–hypochlorite colorimetric method was used for NH_3_-N through a Pentra 400 (Horriba Ltd., Kyoto, Japan) with a calibrated standard of NH_3_-N according to the steps outlined by Rhine et al. [32]. The VFAs were analyzed using a gas chromatograph (GC-2014FRGA1, Shimadzu, Tokyo, Japan) as described by Brotz and Schaefer [33]. The injector temperature was 250 °C, whereas the oven and the detector were maintained at 160 °C. The split injection was 1:25, where N as the carrier gas (flowrate 0.6 mL/min) and make-up gas (flowrate 25 mL/min) were used. The flowrates of hydrogen and air were 20 mL/min and 300 mL/min, respectively.

### 2.7. Statistical Analysis

The datasets were analyzed using the SPSS statistical package (version 22.0; SPSS Inc., Chicago, IL, USA), where animal was used as the experimental unit. The mean value of each variable for each cow was calculated prior to the statistical analyses. The data from all cows were analyzed as a completely randomized design using a generalized linear model procedure, and dietary treatments were considered as fixed effects and the cow as a random effect. The statistical model included the type of diet as an independent variable and observed items such as nutrient intake, digestibility, lactation, etc. as dependent variables. The data were tested for normality using the Anderson–Darling test. A one-way ANOVA was used to detect the effects of diets, including silage type, on nutrient intake and digestibility, milk production, N utilization, and rumen fermentation characteristics. Multiple comparisons of means were determined using Tukey’s post-hoc test. In the current study, all the data were reported as mean values and the standard errors of the mean. Statistical significance was determined at *p* ≤ 0.05, and a statistical tendency was determined at 0.05 < *p* ≤ 0.10.

## 3. Results

### 3.1. Chemical Composition of Silage

MZS had a greater starch content (*p* < 0.01) than HSS and FSS, whereas HSS had a greater WSC content (*p* < 0.01) than MZS and FSS. The CP of MZS (8.07% DM) and HSS (7.56% DM) was greater than that of FSS (6.85% DM, *p* < 0.01; Table 2), while their NDF (*p* = 0.04), ADF (*p* = 0.02), ADL (*p* = 0.04) were less and the pH (*p* = 0.01) lower than those of FSS. The concentrations of total phenolic (*p* = 0.03) and condensed tannin (*p* = 0.04) contents in FSS were greater than those in HSS and MZS.

### 3.2. Milk Production and Efficiency

The yields of actual milk (kg/d, *p* = 0.02), 3.5% FCM (kg/d, *p* = 0.01), milk fat (kg/d, *p* = 0.01), milk protein (kg/d, *p* = 0.02), concentrations of milk fat (%, *p* = 0.01), TS (%, *p* = 0.01), and efficiency of feed-to-milk production (*p* = 0.03 and *p* < 0.01, respectively; Table 3) were greater for cows fed MZS and HSS compared to those fed FSS. The cows fed HSS had the highest production of lactose (1.08 kg/d), TS (2.70 kg/d), and solids-not-fat (1.91 kg/d), followed by those fed MZS, with the lowest values observed for the cows fed FSS (*p* < 0.01, Table 3).

### 3.3. Nutrient Intake and Digestibility

The DM intake (*p* = 0.65), OM (*p* = 0.56), and CP (*p* = 0.63; Table 4) were not affected by the dietary treatments. Intake of NDF was lower in HSS (6.92 kg/d) compared with MZS (7.38 kg/d) and FSS (7.41 kg/d; *p* < 0.01). The cows fed with the FSS had a higher intake of ADF (MZS: 5.29 kg/d, HSS: 5.46 kg/d, and FSS: 5.75 kg/d, *p* < 0.01) and ADL (MZS: 0.98 kg/d, HSS: 0.91 kg/d, and FSS: 1.07 kg/d, *p* = 0.01) than cows fed with MZS and HSS. The highest intake of starch was from MZS (3.88 kg/d, *p* < 0.01), while the highest intake of WSC was from HSS (1.63 kg/d, *p* < 0.01). Both MZS and HSS were associated with greater digestibility of DM (*p* = 0.01), OM (*p* < 0.01), NDF (*p* = 0.01), ADF (*p* = 0.02), and ADL (*p* = 0.03) than FSS.

### 3.4. Nitrogen Utilization

The cows fed FSS experienced increased concentrations of milk urea N (FSS: 8.41 mg/dL, *p* = 0.01), urinary urea N (FSS: 522.86 mg/100 mL, *p* = 0.05), urinary N excretion (FSS: 118.05 g/d, *p* = 0.04), and decreased milk N production (FSS: 103.21 g/d, *p* = 0.02; Table 5) compared with those fed HSS and MZS. MZS and HSS tended to increase the ratio of milk N:N intake (MZS: 0.18, HSS: 0.18, and FSS: 0.17, *p* = 0.06) compared with FSS. No differences were observed in N intake (*p* = 0.63) and N excretion from feces and manure (*p* = 0.52 and 0.67, respectively).

### 3.5. Ruminal Fermentation Characteristics

The cows fed FSS exhibited an increased ruminal pH (MZS: 6.69, HSS: 6.64, and FSS: 6.79, *p* = 0.03; Table 6) and increased NH_3_-N contents (MZS: 13.16 mg/100 mL, HSS: 13.76 mg/100 mL, and FSS: 14.87 mg/100 mL, *p* = 0.02) compared with those fed HSS or MZS. However, acetate (*p* = 0.01) and propionate (*p* = 0.02) concentrations were greater in cows fed MZS and HSS than in those fed FSS. The value of ruminal total VFA (*p* = 0.01) in cows fed HSS (119.52 mmol/L) was within the range observed for cows fed MZS (128.66 mmol/L) and FSS (112.85 mmol/L).

## 4. Discussion

### 4.1. Chemical Composition of Silage

The nutritional characteristics (e.g., DM, CP, fibers, ash, etc.) of forages are closely related to the differences in forage type and agronomic practices. The chemical composition of the silage is a reflection of the different nutrients in the original forages. In the current study, the chemical compositions of MZS, HSS, and FSS were generally consistent with the findings from our previous research [6]. The high levels of WSC in HSS and starch in MZS may be due to the high sugar and maize grains in the pre-ensiling material (high-sugar sorghum and maize) which positively correlated with the residual WSC and starch contents [34]. Greater CP contents in MZS and HSS may be related to the crop CP content prior to ensiling, which might be affected by the type of forage, employed culture system, and method for ensiling. During silage fermentation, a part of the protein can be converted into non-protein N by proteolysis [35]. More residual CP in silage also indicates a less active proteolysis reaction when WSC and starch are greater in forage before ensiling [36].

The greater fiber contents in FSS indicate that there were more cellulose, hemicellulose, and lignin in the FSS cell walls than in MZS and HSS [25]. On the other hand, the fiber content could also be related to the plant’s genotype or the ratio of the leaf blade to the stem [37]. The low levels of fiber in the HSS might be a result of its high WSC content in the plant which is marked by the high content of non-structural carbohydrates that stay in these plants in the form of water-soluble sugars.

### 4.2. Milk Production and Efficiency

Feed quality is the main factor affecting milk yield and milk composition. Under the current study conditions, the cows fed MZS and HSS produced more milk yield and a higher fat concentration in milk than the cows fed FSS (Table 3). This may be because of the amount of nutrient intake and their high digestibility. Greater starch and WSC intake in cows fed MZS-based and HSS-based diets (Table 4) may have increased milk production by increasing the proportion of ruminal propionate (Table 6), a glucogenic precursor [38]. Furthermore, a greater net energy for lactation associated with the MZS and HSS diets rather than with the FSS diet (1.68 Mcal/kg and 1.72 Mcal/kg vs. 1.39 Mcal/kg) may help explain the observed differences in milk yields. A similar finding has been reported, where dairy cows fed diets with lower net energy for lactation produced less milk [39]. This was likely because of the greater fiber intake from the FSS-based diet (Table 4). Intake of indigestible dietary fiber is a major factor that limits the milk yields of dairy cows, particularly in high intakes of ADF and ADL [40].

The observed high milk fat concentration in cows fed with HSS and MZS when compared with those fed FSS in this study are inconsistent with the results from other studies. Malekkhahi et al. [41] reported that the milk fat concentration was reduced when the cows received high-starch compared with low-starch diets. However, milk fat concentration was not affected in other studies by Boerman et al. [42] and Ran et al. [43], implying that additional factors might have been involved. In addition, the content and proportion of lignin in the cell wall are important factors affecting milk fat and milk yield. The reasons for milk fat differences lie possibly in differences in chemical composition, nutrient utilization, and the characteristics of silage crops that can be affected by differences in agronomic practices and cultivars. The fat concentration in milk is also influenced by factors such as milk yield and milk composition (Table 3).

The greater yields of 3.5% FCM and TS could be explained by the high fat concentration in the milk of cows fed with either HSS or MZS compared with those fed FSS. Consistent with the present findings, Yang et al. [44] demonstrated that silage type (MZS vs. HSS) did not affect milk protein or lactose concentrations, whereas Sánchez-Duarte et al. [45] found that the dairy cows fed brown midrib sweet sorghum silage experienced reduced milk protein concentrations compared with cows fed with the conventional sweet sorghum silage or MZS-based diets. This may be due to the differences in nutrient contents among different sorghum varieties.

### 4.3. Nutrient Intake and Digestibility

The cows fed with FSS-based diets had greater fiber intake, less starch, and a lower WSC intake. This was a result of the feeding of greater proportions of dietary fiber and less starch, as well as WSC, compared with those cows fed diets containing MZS and FSS (Table 2). However, no differences were observed in the intake of DM, OM, and CP, a phenomenon which may reflect the similar palatability of the experimental diets. The results are consistent with the study by Oliver et al. [15], in which the cows fed with sorghum silage diets consumed similar DM compared to those fed maize silage diets. That is because of the high residual WSC in the HSS which is beneficial to increasing silage quality and palatability.

The forage lignin–cellulose complex, especially lignin, is one of the main obstacles to the utilization of nutrients, as it absorbs enzymes and decreases the accessibility of digestive enzymes to cellulose and xylan, resulting in lower rates of fiber hydrolysis [46]. Lignin cannot be degraded under ruminal anaerobic conditions and reduces fiber digestibility by shielding the cell wall carbohydrates [47,48]. Therefore, the greater concentration of fibers, including lignin, in FSS compared to MZS and HSS may be a reason for the lower digestibility of the FSS-based diet compared with MZS- and HSS-based diets (Table 2). Likewise, previous research [6] also confirmed that HSS was associated with greater in vitro digestibility of DM and OM than FSS. Additionally, lower nutrient digestibility in the FSS-based diet may be due to the greater concentrations of phenolic and tannin compounds in FSS than in MZS and HSS. Phenolic compounds and tannins have been confirmed to prevent dietary protein and structural carbohydrates from being degraded by ruminal microorganisms [49].

### 4.4. Nitrogen Utilization

Overall, the N utilization data from this study showed that feeding MZS and HSS with high degradability of nutrients resulted in greater N utilization, with lower contents of milk urea N, urinary urea N, and urinary N excretion (Table 5). The negative relationship between N utilization efficiency and milk urea N and a positive relationship between N utilization efficiency and WSC have been described previously [50]. Increased dietary WSC content could correct the imbalance between the energy and protein supply in the rumen and improve the capture of ruminal ammonia in the form of microbial protein, thereby influencing the supply of absorbed amino acids for milk protein production [51]. The better N utilization from cows fed MZS and HSS might be a result of a greater efficiency in turning feed N into milk N. The greater contents of fiber, tannins, and phenolic compounds were also reported to be relevant to the prevention of dietary starch and protein degradation by inhibiting rumen microbial enzymes [49,52], resulting in lower digestibility of protein with greater NH_3_-N levels [53]. This could lead to lower N use efficiency in an FSS-based diet. MZS and HSS tended to increase the ratio of milk N to N intake and milk N to manure N compared with FSS, primarily because cows fed with MZS and HSS excreted more N into milk, a phenomenon which is beneficial to prevent environmental pollution from dairy farming.

### 4.5. Ruminal Fermentation Characteristics

The great proportions of fermentable carbohydrates such as starch and WSC might have contributed to the lower ruminal pH and NH_3_-N in cows fed with MZS and HSS compared to those fed with FSS. The ruminal NH_3_-N content could be greatly affected by diet, feeding time, feeding interval, and animal factors [54]. The availability of carbohydrates is a primary factor in promoting the efficiency of NH_3_-N production and dietary N utilization in ruminants [55,56]. Thus, the lower pH and NH_3_-N, probably, were a result of less proteolysis occurring, and more readily fermentable carbohydrates were available from MZS and HSS than from FSS during their rumen fermentation. Owens et al. [57] also observed lower concentrations of rumen NH_3_-N in cattle fed maize silage compared to cattle fed either grass silage, fermented whole-crop wheat, or urea-processed whole-crop wheat. Similarly, some studies have also reported that supplementation with starch, sucrose, or fructose for sheep or cattle reduced rumen pH and NH_3_-N levels [58,59,60].

Ruminal VFA is the primary source of energy for ruminants. The ruminal VFA type varies according to the fermented substrates, including starches and soluble sugars, the microbial population, and ruminal situation [61]. The present findings of increased concentrations of total ruminal VFA, acetate, and propionate in the cows fed MZS and HSS are likely due to the faster microbial fermentation of starch in MZS and WSC in HSS, as well as to the fiber types in MZS and HSS that were more fermentable than those in FSS, providing an appropriate fermentation environment for the proliferation of microbes.

## 5. Conclusions

Higher proportions of starch in MZS and WSC in HSS were observed, as expected. MZS and HSS had greater CP but a lower pH and fiber concentration than FSS. Compared with the FSS-based diet, both the MZS- and the HSS-based diets increased nutrient digestibility, milk yield, feed conversion efficiency, N use efficiency, total ruminal VFA, acetate, and propionate, but decreased ruminal NH_3_-N concentrations and pH for dairy cows. Based on the current results, this study indicates that replacing MZS with HSS, without additional grain supplements, has no negative influences on feed intake, milk yield, N utilization, or ruminal fermentation.

## Figures and Tables

**Table 1 animals-14-01702-t001:** Ingredients and chemical composition of experimental diets offered to cows.

Component	Experimental Silage in the Diet		
	MZS	HSS	FSS
Ingredient, % of DM			
Maize silage	40	-	-
High-sugar sorghum silage	-	40	-
Forage sorghum silage	-	-	40
Wheat straw meal	3	3	3
Alfalfa hay meal	18	18	18
Cottonseed hull	9	9	9
Cottonseed meal	18	18	18
Maize grain meal	10	10	10
Vitamin–mineral mix ^1^	1	1	1
Limestone	0.2	0.2	0.2
Salt	0.3	0.3	0.3
Sodium bicarbonate	0.5	0.5	0.5
Chemical composition, % of DM			
DM	66.08	65.79	66.13
CP	16.17	15.97	15.68
OM	92.31	92.29	92.05
NDF	30.93	29.23	31.59
ADF	22.20	22.23	24.71
ADL	4.50	3.84	5.37
EE	2.48	2.31	2.39
WSC	3.52	6.95	3.08
Starch	15.14	11.34	11.29
Ash	8.05	8.09	8.32
Ca	0.50	0.49	0.49
P	0.39	0.37	0.37
NFC ^2^	43.37	44.40	42.02
NE_L_ ^3^, Mcal/kg	1.68	1.72	1.39

Abbreviations: MZS = maize silage; HSS = high-sugar sorghum silage; FSS = forage sorghum silage; DM = dry matter; CP = crude protein; OM = organic matter; NDF = neutral detergent fiber; ADF = acid detergent fiber; ADL = acid detergent lignin; EE = ether extract; WSC = water-soluble carbohydrate; NFC = non-fiber carbohydrates; NE_L_ = net energy for lactation. ^1^ Provided per kg of DM: 162 g of Ca, 42 g of Mg, 8 g of S, 330 mg of Se, 44,537 mg of Zn, 46,635 mg of Mn, 22 mg of Fe, 12 mg of I, 8376 mg of Cu, 550 mg of Co, 53,557 IU of vitamin A, 12,787 IU of vitamin D, and 3145 IU of vitamin E (Tecon Husbandary, Akesu, Xinjiang, China). ^2^ NFC = 100 − (CP % + NDF % + EE % + Ash %), using tabular values for EE [20]. ^3^ NE_L_ based on tabular values [20].

**Table 2 animals-14-01702-t002:** Chemical composition of maize silage (MZS), high-sugar sorghum silage (HSS), and forage sorghum silage (FSS).

Items	Experimental Silage			SEM	*p*-Value
	MZS	HSS	FSS		
DM, %	29.45	28.72	29.56	0.25	0.56
CP, %DM	8.07 ^a^	7.56 ^a^	6.85 ^b^	0.35	<0.01
EE, %DM	2.36	2.32	2.34	0.12	0.18
WSC, %DM	4.76 ^b^	13.34 ^a^	3.65 ^c^	0.01	<0.01
NDF, %DM	38.81 ^b^	34.56 ^c^	40.45 ^a^	1.52	0.04
ADF, %DM	26.45 ^b^	27.78 ^b^	31.65 ^a^	1.12	0.02
ADL, %DM	4.39 ^b^	3.23 ^b^	4.57 ^a^	0.11	0.04
Starch, %DM	12.87 ^a^	6.87 ^b^	6.76 ^b^	0.21	<0.01
Ash, %DM	4.85	4.95	5.14	0.08	0.06
Total phenolic, %DM	1.37 ^b^	1.41 ^b^	1.53 ^a^	0.74	0.03
Condensed tannin, %DM	1.02 ^c^	1.17 ^b^	1.35 ^a^	0.89	0.04
NH_3_-N, mg/100 mL	6.42	5.33	5.21	0.12	0.07
pH	3.84 ^b^	3.79 ^b^	4.52 ^a^	0.07	0.01

Abbreviations: MZS = maize silage; HSS = high-sugar sorghum silage; FSS = forage sorghum silage; DM = dry matter; CP = crude protein; EE = ether extract; WSC = water-soluble carbohydrates; NDF = neutral detergent fiber; ADF = acid detergent fiber; ADL = acid detergent lignin; NH_3_-N = ammonia-nitrogen. ^a–c^ In the same row, values with different superscripts were significantly different (*p* < 0.05).

**Table 3 animals-14-01702-t003:** Milk production, milk composition, and efficiency of lactating cows fed diets including maize silage (MZS), high-sugar sorghum silage (HSS), and forage sorghum silage (FSS).

Items	Experimental Silage in the Diet			SEM	*p*-Value
	MZS	HSS	FSS		
Milk yield, kg/d					
Milk	21.24 ^a^	21.54 ^a^	20.29 ^b^	0.19	0.02
3.5% FCM ^1^	21.86 ^a^	22.16 ^a^	20.64 ^b^	0.21	0.01
Milk composition, %					
Fat	3.68 ^a^	3.68 ^a^	3.61 ^b^	0.02	0.01
True protein	3.18	3.12	3.08	0.13	0.37
Lactose	4.94	5.04	4.93	0.02	0.11
TS	12.50 ^a^	12.54 ^a^	12.34 ^b^	0.04	0.01
SNF	8.82	8.86	8.72	0.04	0.16
Milk component yield, kg/d					
Fat	0.78 ^a^	0.79 ^a^	0.73 ^b^	0.01	0.01
True protein	0.67 ^a^	0.67 ^a^	0.62 ^b^	0.01	0.02
Lactose	1.05 ^b^	1.08 ^a^	1.01 ^c^	0.01	<0.01
TS	2.65 ^b^	2.70 ^a^	2.51 ^c^	0.03	<0.01
SNF	1.87 ^b^	1.91 ^a^	1.77 ^b^	0.02	<0.01
Efficiency					
Milk yield/DMI	0.91 ^a^	0.92 ^a^	0.87 ^b^	0.01	0.03
3.5% FCM/DMI	0.92 ^b^	0.96 ^a^	0.88 ^c^	0.01	<0.01

Abbreviations: MZS = maize silage; HSS = high-sugar sorghum silage; FSS = forage sorghum silage; FCM = fat-corrected milk; TS = total solid; SNF = solids-not-fat; DMI = DM intake. ^1^ 3.5% FCM = (0.4324 × milk yield, kg/d) + (16.216 × milk fat yield, kg/d). ^a–c^ In the same row, values with different superscripts were significantly different (*p* < 0.05).

**Table 4 animals-14-01702-t004:** Nutrient intake and digestibility of lactating cows fed diets including maize silage (MZS), high-sugar sorghum silage (HSS), and forage sorghum silage (FSS).

Items	Experimental Silage in the Diet			SEM	*p*-Value
	MZS	HSS	FSS		
Intake, kg/d					
DM	23.71	23.51	23.28	0.18	0.65
OM	21.89	21.68	21.43	0.17	0.56
CP	3.84	3.79	3.77	0.03	0.63
NDF	7.38 ^a^	6.92 ^b^	7.41 ^a^	0.06	<0.01
ADF	5.29 ^b^	5.46 ^b^	5.75 ^a^	0.05	<0.01
ADL	0.98 ^b^	0.91 ^b^	1.07 ^a^	0.01	0.01
WSC	0.84 ^b^	1.63 ^a^	0.71 ^c^	0.05	<0.01
Starch	3.88 ^a^	2.73 ^b^	2.63 ^b^	0.03	<0.01
Digestibility, %					
DM	65.35 ^a^	63.48 ^a^	58.65 ^b^	1.57	0.01
OM	66.41 ^a^	65.89 ^a^	62.10 ^b^	1.28	<0.01
CP	74.21	75.34	74.01	1.64	0.32
NDF	57.62 ^a^	56.21 ^a^	51.30 ^b^	2.10	0.01
ADF	54.32 ^a^	53.21 ^a^	50.14 ^b^	1.89	0.02
ADL	13.57 ^a^	13.32 ^a^	10.26 ^b^	2.52	0.03
WSC	95.63	96.32	96.65	3.25	0.57
Starch	97.63	98.45	97.56	2.12	0.32

Abbreviations: MZS = maize silage; HSS = high-sugar sorghum silage; FSS = forage sorghum silage; DM = dry matter; OM = organic matter; CP = crude protein; NDF = neutral detergent fiber; ADF = acid detergent fiber; ADL = acid detergent lignin; WSC = water-soluble carbohydrates. ^a–c^ In the same row, values with different superscripts were significantly different (*p* < 0.05).

**Table 5 animals-14-01702-t005:** Nitrogen utilization of lactating cows fed diets including maize silage (MZS), high-sugar sorghum silage (HSS), and forage sorghum silage (FSS).

Items	Experimental Silage in the Diet			SEM	*p*-Value
	MZS	HSS	FSS		
N balance					
N intake, g/d	613.79	606.77	602.41	4.81	0.63
Milk N, g/d	107.20 ^a^	109.55 ^a^	103.21 ^b^	0.92	0.02
MUN, mg/dL	7.80 ^b^	7.58 ^b^	8.41 ^a^	0.12	0.01
Urinary urea N, mg/100 mL	514.92 ^b^	501.13 ^b^	522.86 ^a^	4.09	0.05
Urinary N excretion ^1^, g/d	111.72 ^b^	107.86 ^b^	118.05 ^a^	1.71	0.04
Fecal N excretion ^2^, g/d	394.88	389.36	381.84	4.61	0.52
Manure N excretion ^3^, g/d	506.60	497.22	499.19	4.59	0.67
N utilization efficiency					
Milk N:N intake ^4^	0.18	0.18	0.17	0.01	0.06
Milk N:manure N ^5^	0.21	0.22	0.20	0.02	0.09

Abbreviations: MZS = maize silage; HSS = high-sugar sorghum silage; FSS = forage sorghum silage; N = nitrogen; MUN= milk urea nitrogen. ^1^ Predicted using the equation: 0.026 × MUN, mg/100 mL × BW, kg [31]. ^2^ Predicted using the equation: N intake, g/d—urinary N excretion, g/d—milk N, g/d. ^3^ Manure N, g/d = urinary N excretion, g/d + fecal N excretion, g/d. ^4^ Milk N:N intake = ratio of milk N to N intake. ^5^ Milk N:manure N = ratio of milk N to manure N. ^a,b^ In the same row, values with different superscripts were significantly different (*p* < 0.05).

**Table 6 animals-14-01702-t006:** Ruminal fermentation characteristics of lactating cows fed diets including maize silage (MZS), high-sugar sorghum silage (HSS), and forage sorghum silage (FSS).

Items	Experimental Silage in the Diet			SEM	*p*-Value
	MZS	HSS	FSS		
pH	6.69 ^b^	6.64 ^b^	6.79 ^a^	0.11	0.03
NH_3_-N, mg/100 mL	13.16 ^b^	13.76 ^b^	14.87 ^a^	0.43	0.02
total VFA, mmol/L	128.66 ^a^	119.52 ^ab^	112.85 ^b^	3.76	0.01
Acetate, mmol/L	84.46 ^a^	85.58 ^a^	76.94 ^b^	2.67	0.01
Propionate, mmol/L	23.37 ^a^	22.76 ^a^	18.68 ^b^	1.16	0.02
Butyrate, mmol/L	15.59	11.89	11.03	0.17	0.08
Iso-butyrate, mmol/L	2.24	2.27	2.33	0.04	0.55
Valerate, mmol/L	0.99	1.02	0.16	0.11	0.08
Iso-valerate, mmol/L	1.05	1.14	1.18	0.09	0.16
Acetate:propionate	3.10	3.08	3.02	0.17	0.27

Abbreviations: MZS = maize silage; HSS = high-sugar sorghum silage; FSS = forage sorghum silage; NH_3_-N = ammonia-nitrogen; VFA = volatile fatty acids. ^a,b^ In the same row, values with different superscripts were significantly different (*p* < 0.05).

## Data Availability

No data were deposited in an official repository. Upon reasonable request, the data and statistical models are available from the corresponding author.

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
