# Peer review of "Effects of Sweet and Forge Sorghum Silages Compared to Maize Silage without Additional Grain Supplement on Lactation Performance and Digestibility of Lactating Dairy Cows"

_animals, 2024, doi:10.3390/ani14111702_

Round 1

Reviewer 1 Report

Comments and Suggestions for Authors

The title of the manuscript is ambiguous and contradictory to the content of the text.

abstract:

Instead of using the term 'fiber', it would be preferable to present the results of NDF and ADF.

The results of rumen fermentation (acetate, propionate, etc.) are not included in the abstract.

Materials and Methods:

The number of animals in each treatment group is small, which can affect the statistical power.

The duration of the experiment is short.

Particle sizes and results for Penn State Particle Size Separator (PSPS) regarding MZS, HSS, and FSS, as well as all three diet, were not reported in the paper.

Why are mean comparisons done using Duncan's ? It is suggested to compare means using Tukey's.

In Table 2, the starch level of maize silage is not suitable.

How was the state of rumination behavior? These results must be reported.

In the discussion section, some of the results have been repeated and need to be rewritten.

In general, more data are needed to properly evaluate this paper.

Comments on the Quality of English Language

.

Reviewer 2 Report

Comments and Suggestions for Authors

Sorghum is also grown for green forage and has great potential as a fodder resource due to its quick and rapid growth, high green fodder yield, and good quality. The whole plant is often used as forage, hay, and silage. Forage sorghum hybrids and varieties are gaining prominence worldwide in place of corn because of their low water requirement. Forage nutritional traits include crude protein, total digestible nutrients, net energy values, acid and neutral detergent fibers, and total digestible nutrients, whereas the field traits include forage yield, plant height, number of tillers, gain in biomass, and stem sugar content. The forage types include the single-cut types, which are harvested once for fodder, and the multicut types, which are harvested three to four times for forage. Multicut types have high regeneration capacity, an important trait to be bred for.

Utilization of properly processed sorghum grain in lactating dairy cow diets results in similar milk production as corn. This has been proven in numerous research trials. Processing is very important to get maximum utilization of the sorghum. Steam-flaking can increase the energy value of sorghum 13 to 20 percent and increases the utilization of sorghum in the rumen to produce microbial protein. These increases are a result of the disruption of the starch matrix and the protein matrix covering the starch due to the moisture, heat and pressure associated with steam-flaking. This results in increased milk production as compared to dry-rolling. Steam-flaked sorghum results in similar milk production as steam-flaked corn. Any grain or starch can be utilized to produce ethanol. Sorghum is the second most common grain used to produce ethanol in the U.S., with corn being the primary feedstock. The resulting sorghum byproduct can be used in dairy cattle diets. Inconsistent results in animal feeding trials may be due to different processes used at the ethanol plant. New ethanol technologies, such as oil recovery, may further add to a nutritionally inconsistent byproduct and contribute to varying results observed in animal feeding trials.

Taking into account the above information, I think that so far research on this topic has been done quite often and with high quality, and therefore the sentence in line 24-25 must be softened or recomposed (for example, in vitro studies have been done so far, and we will do in vivo study). You should also describe in a little more detail how and why the grains were added and why you think they won't be needed in your experimental conditions.

The abstract is well written with sufficient information.

In the introduction, sufficient information is given about previous research and the reasons for conducting the current research. The last paragraph of the introduction reflects the originality of the research approach, because the authors try to prove that "sugar in sweet sorghum silages can provide the required proportion of starch, which can maintain the milk production performance of dairy cows

and improve the efficiency of N-utilization" (line 101-102).

The design and all the information offered in MandM are adequate.

Question regarding Table 1: are the meal results obtained related to tabular calculations or are the data obtained in some other way? Add the information to line 145.

All experimental procedures related to animals and sampling are clear and allow a high level of reproducibility of the experiment.

The results are clear and unambiguous.

The discussion is well organized and follows the results. The authors have well described the physiological processes behind this kind of experimental diet, emphasizing the propionate and fatty acid composition of the rumen, on which digestibility and milk production depend. Overall the discussion is well written.

The main result and assumption are drawn through the entire manuscript and at the end the authors point out in the very conclusion, which is that "Based on the current results, this study indicated that replacing MZS with HSS, without additional grain supplements, had no negative influences on feed intake, milk yield, N utilization, or ruminal fermentation."

Although this assumption and this finding will be somewhat of a surprise to many authors, I think that the manuscript should be published in order to enable further research in this area and with this experimental setup.

Reviewer 3 Report

Comments and Suggestions for Authors

General information:

-        Many cow farmers in dry areas suffer from a lack of feed. By using forage sorghum (FS) or high sugar sorghum (HS) for silage instead of maize silage, we can reduce this feed gab.

-        To obtain accurate results, it would have been better to conduct preliminary tests. I mean using 3 dairy cows with fistulas to check the digestibility and the performance. After the initial results, you set up a large-scale experiment. Nevertheless, the results showed a possibility of using such feedstuffs.

Some comments on the manuscript:

-        Regarding the materials and methods, I noticed that the authors did not mention the health status of the test dairy cows during the experiment, even though this factor plays a very important role in the animal's performance. I mean, when the cows are sick (subclinical mastitis), the animals milk production decreases and the milk quality deteriorate without being noticed from outside. The question is whether an udder health control measure such as the California mastitis test or the determination of somatic cells in the milk produced was carried out?

-        In line 175 the fat-corrected milk of 3.5% (FCM) according to Holt et al (2013) was calculated. The question is, why did not you use an equation from China? We know that each breed has a certain characteristic that distinguishes it from other breeds, primarily the ratio between protein and fat in the milk.

-        In line 179 it was mentioned that the milk ingredients were determined using an infrared device. Is the device calibrated for total nitrogen (total protein) or protein nitrogen (true protein) in milk?

-        Line 190 stated that the faecal and feed samples were taken in period 36 to 38. The question arises as to whether the interval period from feed intake to fecal collection is considered?

-        Line 194 stated that the fecal samples were stored for chemical analysis. However, the authors did not mention how and at what temperature they were stored.

-        On line 197, an equation was written without describing the meaning of the indicators.

-        In lines 394 and 395, the increase in dry matter in HSS and MZS groups compared to the FSS group was explained by a high concentration of fat in the milk. This is incorrect because other ingredients also play a role, as table 3 shows.

-        On lines 541 and 542, reference 19 is a report or what and where is the page number?  
